# LLM Embeddings Improve Test-time Adaptation to Tabular $Y|X$-Shifts

## Abstract

For tabular datasets, the change in the relationship between the label and covariates ($Y|X$-shifts) is common due to missing variables. Since it is impossible to generalize to a completely new and unknown domain, we study models that are easy to adapt to the target domain even with few labeled examples. We focus on building more informative representations of tabular data that can mitigate $Y|X$-shifts, and propose to leverage the prior world knowledge in LLMs by serializing the tabular data to encode it. We find LLM embeddings alone provide inconsistent improvements in robustness, but models trained on them can be well adapted to the target domain even using 32 labeled observations. Our finding is based on a systematic study consisting of 7650 source-target pairs and benchmark against **261,000** model configurations trained by 20 algorithms. Our observation holds when ablating the size of accessible target data and different adaptation strategies.

## 1 Introduction

Predictive performance degrades when the distribution of target domain shifts from that of source (training) (Bandi et al., 2018; Wong et al., 2021; Hand, 2006; Ding et al., 2021a; Amorim et al., 2018). Distribution shifts can be categorized into shifts in the marginal distribution of covariates ($X$-shifts) or changes in the relationship between the label and covariates ($Y|X$-shifts). In computer vision, $X$-shifts are prevalent since high-quality human labels are consistent across different images (Recht et al., 2019; Miller et al., 2021; Shankar et al., 2019); in contrast, $Y|X$-shifts are prevalent in tabular data due to missing variables and hidden confounders. There is a large body of work addressing $X$-shifts due to its dominance in vision and language (Li et al., 2017; Zhuang et al., 2020; Zhou et al., 2022), yet the work on $Y|X$-shifts remain relatively limited (Liu et al., 2023).

The main challenge with addressing $Y|X$-shifts in tabular tasks is that the source data may provide little insight on the target distribution. Since it is impossible to generalize to a completely new and unknown domain (Arjovsky et al., 2019; Rosenfeld et al., 2021), we focus on leveraging few labeled target examples (on the order of 10 to 100) to address small $Y|X$-shifts that negatively impact model performance. Our goal is to build a feature representation $\phi(X)$ such that the difference between $\mathbb{E}_{\text{source}}[Y|\phi(X)]$ and $\mathbb{E}_{\text{target}}[Y|\phi(X)]$ are learnable even based on a few target data.

Using the wealth of world knowledge learned during pre-training, LLMs have the potential to build representations that mitigate the impact of confounders whose distribution changes across source and target. Specifically, we use a LLM encoder (`e5-Mistral-7B-Instruct`) to featurize tabular data—which we referred to as LLM embeddings—and fit a shallow neural network (NN) on these embeddings for tabular prediction (Figure 1). In contrast to classical numerical encoding of tabular data, our approach automatically incorporates the semantics of each covariate using off-the-shelf LLMs, and can include additional contextual domain-level information that can help account for missing variables whose distribution shifts from source to target.

In this work, we restrict attention to *lightweight* probing approaches instead of expensive end-to-end updates (Hegselmann et al., 2023; Yang et al., 2024) that require full weight access and weight updates to the LLM (we find this infeasible across the thousands of source-target pairs we consider). Throughout our investigation, we use the same LLM encoder to extract the LLM embeddings, and we only "finetune" the shallow NNs we use as the main prediction model across target domains. Investigating how different LLM encoders affect tabular $Y|X$ shifts is left as a future work.

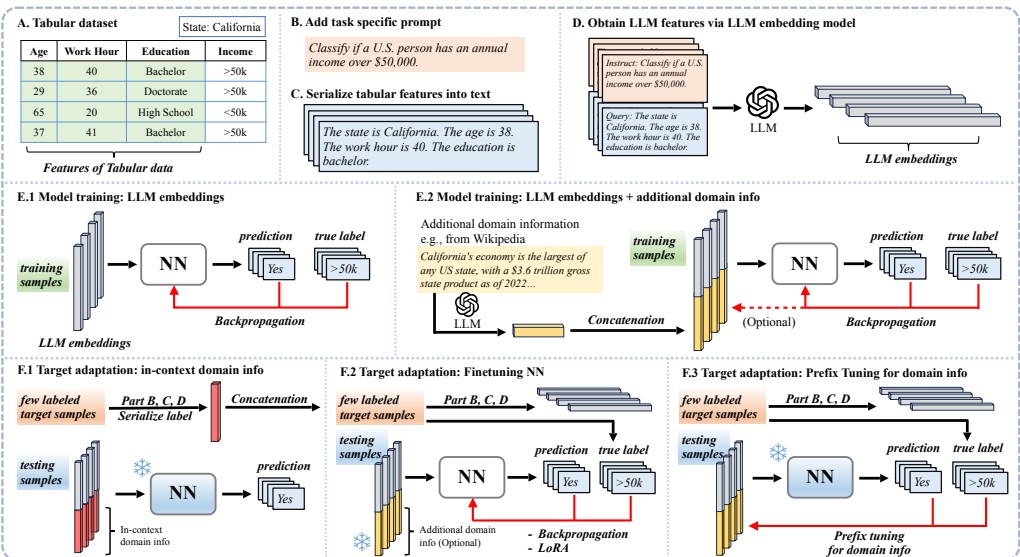

Figure 1: Overview of methods incorporating LLM embeddings.

For rigorous empirical evaluation, we consider **7,650** natural spatial shifts (source→target) based on three real-world tabular datasets (ACS Income, Mobility, Pub.Cov (Ding et al., 2021a)). Our testbed serves as a *large-scale* benchmark for $Y|X$-shifts on tabular data, offering a standardized protocol for training, validation, testing, and finetuning, as well as a consistent hyperparameter selection process. Compared to previous benchmarks on tabular distribution shifts (Liu et al., 2023), this paper not only explores a significantly greater variety of shift settings but also introduces a series of novel approaches to incorporate LLM embeddings as features. Such a comprehensive evaluation ensures the robustness and adaptability of our findings across diverse scenarios, setting a new standard for future work in this domain. We compare our proposed approach with typical methods on Tabular features, including basic models (LR, SVM, NN), gradient-boosting trees (GBDT; XGB, LGBM, GBM), and distributionally robust methods (DRO; KL-DRO, $\chi^2$-DRO, Wasserstein DRO, CVaR-DRO, and Unified-DRO). In total, we consider 20 algorithms and **261,000** model configurations.

Since it is unrealistic to expect any single method to uniformly dominate over large number of source→target settings, we complement traditional average-case metrics using the *fraction of times each method performs best*. For each method $\mathcal{M}$,

$$\text{FractionBest}(\mathcal{M}; \Delta) := \frac{|\mathcal{S}(\Delta) \cap \mathcal{S}_{\mathcal{M}}|}{|\mathcal{S}(\Delta)|}, \tag{1}$$

where $\mathcal{S}(\Delta)$ contains all source→target settings where the performance between the best and second best model is larger than $\Delta$ (we set $\Delta = 1\%$ in this paper), and $\mathcal{S}_{\mathcal{M}}$ contains all source→target settings where model $\mathcal{M}$ performs the best. FractionBest calculates the proportion of source→target settings where (i) $\mathcal{M}$ outperforms all other methods and (ii) the improvement over the second-best model is meaningful (and significant).

First, we consider LLM embeddings without any adaptation to labeled target data[1]. Shallow networks based on LLM embeddings (LLM|NN) outperform all other methods on tabular features in **85%** settings in the ACS Income dataset, and in **78%** in the ACS Mobility dataset. However, for the ACS Pub.Cov dataset, the FractionBest drops to **44%**, which indicates that LLM embeddings do not always offer a perfect solution (see Figure 2 (a)-(c)). We conclude *LLM embeddings sometimes improve robustness*, but *do not consistently surpass* state-of-the-art tree-ensemble methods.

However, we find that *finetuning* the prediction model (shallow NN) *with few target samples* can *make a big difference* even when using identical LLM embeddings. When finetuning with just 32 target samples, the FractionBest ratio (Equation (1)) remains at **85%** on ACS Income, improves from 78% to **86%** on ACS Mobility, and from 44% to **56%** on ACS Pub.Cov (see Figure 2 (d)-(f)). We find this improvement *surprising*: although the shallow NN has numerous parameters,

---

[1]We do not conduct any target adaptation; however, we use 32 labeled target samples for validation (selection of hyperparameters, etc.).

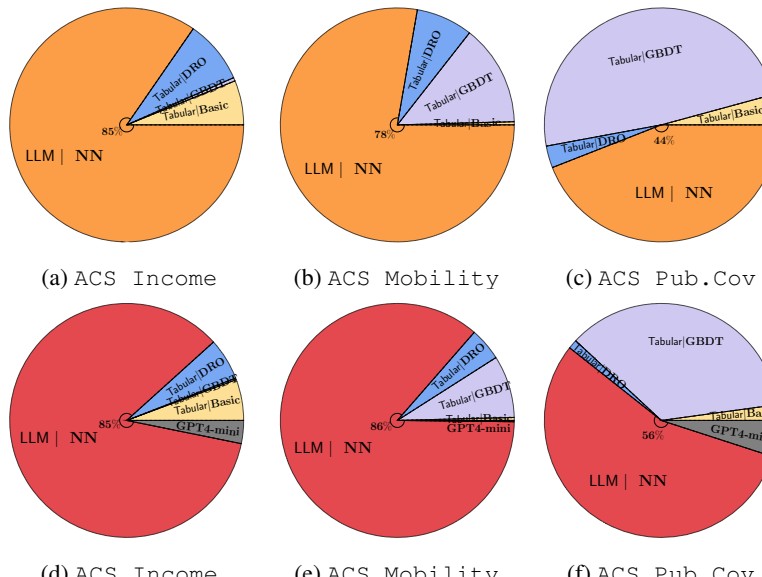

Figure 2: The FractionBest Ratio in Equation (1) (with $\Delta = 1\%$). We compare our proposed methods—(a)-(c): LLM|NN and (d)-(f): LLM|NN (finetuning)—with methods on Tabular features.

finetuning with only 32 target samples surprisingly improve target performance by a relatively large margin. More importantly, such improvement is observed under $Y|X$ shifts and holds across a wide range of distributional shift settings. This implies the potential of our novel and lightweight approach, and opens up the door for further investigation of using LLM embeddings in tabular classification tasks. Theoretical insights are discussed in Section 4.

Our method also implies multiple additional benefits (See Section 3.2). With the same amount of target samples, the finetuned NNs significantly outperform in-context learning with GPT4-mini, the SOTA decoder model (see Figure 2 (d)-(f)). Moreover, *the performance gain brought by target samples is larger under stronger $Y|X$-shifts*, where the level of $Y|X$-shifts is measured by DISDE (Cai et al., 2023). Finetuning with 32 target samples yields an average performance gain of 5.4 percentage points across the worst 500 settings on ACS Pub.Cov, compared to no finetuning. This is notably higher than the 1.2% average gain observed across all 2550 settings (**4.5 times**).

Beyond our primary findings, we also conduct ablation studies to better understand our approach in Section 3.3. Given the large number of model parameters and limited labeled target samples, one might expect parameter-efficient methods like Low-Rank Adaptation (LoRA) (Hu et al.) and Prefix Tuning (Li & Liang, 2021) to offer a clear advantage for target adaptation. However, we find *the specific finetuning approach has small impact* on target adaptability under tabular $Y|X$-shifts. In Figure 7, all target adaptation methods significantly outperform the non-finetuned version when using LLM embeddings. On the other hand, *incorporating the "right" domain information has an outsize impact* on adaptability to $Y|X$-shifts. For ACS Pub.Cov, adding additional domain information from Wikipedia shows minimal improvement alone, but significantly enhances performance under target adaptation.

Another practical question is how to allocate a fixed number of labeled target samples between target adaptation and validation (selection of finetuning method, hyperparameters, etc). In Figure 8 to come, we compare two allocation schemes of 64 labeled target samples: (i) using all 64 samples for validation (solid bar), and (ii) dividing the samples into 32 for validation and 32 for finetuning (shaded bar). For ACS Mobility and ACS Pub.Cov, *target adaptation provides significant gains over the validation-only approach*, highlighting the need for further investigation into sample allocation.

**Related work** Tabular data is a common modality in electronic health records, finance, and social and natural sciences. Unlike other modalities like images and text, gradient-boosted trees (GBDT; GBM (Friedman, 2001; 2002), XGBoost (Chen & Guestrin, 2016), LGBM (Ke et al., 2017)) remain the state-of-the-art (Gorishniy et al., 2021; Shwartz-Ziv & Armon, 2022; Gardner et al., 2022) even when compared to neural networks specifically designed for tabular data (Arik & Pfister, 2021; Huang et al., 2020; Kadra et al., 2021; Katzir et al., 2020). GBDTs have recently been observed to provide strong performance under distribution shifts, which forms the basis of its use as a main

baseline (Gardner et al., 2022; Liu et al., 2023). In addition, some recent works employ in-context learning to address the few-shot classification problem (Hegselmann et al., 2023; Yang et al., 2024). However, they are significantly more resource-intensive to finetune the LLM itself. Given the huge number of settings, we only compare with GPT4-mini for this line of research in this work. Besides, while we use standard LLMs to generate embeddings, LLMs specialized for tabular data (Yan et al.) can also be used as embedding extractors.

A wide range of methods have been proposed to address distribution shifts, notably robust learning methods, balancing methods, and invariant learning methods. Distributionally robust optimization (DRO) construct an uncertainty set around the training distribution and optimizing for the worst-case distribution within this set, thereby mitigating the impact of potential distribution shifts. Variants of DRO methods have been developed using different distance metrics, such as $\chi^2$-divergence (Duchi & Namkoong, 2021; Duchi et al., 2021), KL-divergence (Hu & Hong, 2013), and Wasserstein distance (Blanchet et al., 2017; 2018; 2019a; 2023a). However, these approaches have recently been observed to be ineffective in addressing real-world tabular distribution shifts Liu et al. (2023). On the other hand, invariant learning (Peters et al., 2016; Arjovsky et al., 2019; Koyama & Yamaguchi, 2020) seeks to learn causally invariant relationships across multiple *pre-defined* environments. In this work, we include 5 typical DRO methods but do not consider invariant learning methods as they require multiple training environments, which is not the focus of this work. Some statistical works (Li et al., 2022; Tian & Feng, 2023) provide theoretical guarantees for simple linear models in transfer learning, but these guarantees often do not extend to more complex models like decision trees or neural networks used in real-world applications. This version improves clarity and flow while retaining the original meaning. Additionally, many studies have explored domain adaptation (Iwasawa & Matsuo, 2021; Liang et al., 2023; Chen et al., 2023). However, most of these focus on $X$-shifts in image data, whereas our work addresses $Y|X$-shifts in tabular data.

## 2 METHODS

In this section, we introduce a series of methods utilizing LLM embeddings for tabular prediction, as well as different choices to incorporate additional domain information, different model architectures, and target adaptation techniques using a small amount of labeled target samples. To the best of our knowledge, this work is the first to comprehensively explore the impact of LLM embeddings on tabular $Y|X$-shifts.

### 2.1 LLM EMBEDDINGS FOR TABULAR PREDICTION

We first introduce how we transform tabular data into LLM embeddings, where the key idea is to serialize each sample into a natural language format that the LLM can process. There is a substantial body of research on serialization, including using another LLM to rewrite tabular data into natural language (Hegselmann et al., 2023), adding descriptions of the classification task, training and test examples (Hegselmann et al., 2023; Slack & Singh, 2023), etc. Among these methods, Hegselmann et al. (2023) demonstrate that using a straightforward text template with a task description consistently achieves the best empirical performance.

Using an income prediction problem to illustrate, consider a simple task description such as "`Classify whether US working adults' yearly income is above $50000 in 2018.`" along with a simple serialization template that enumerates all features in the format "`The [feature name] is [value]`". Adopting this serialization approach, we employ the encoder model `e5-Mistral-7B-Instruct` to generate the LLM embedding. Formally, the encoder takes the serialization Serialize($X$) of sample $X$ as input and outputs its corresponding embedding $\Phi(X)$ as

$$X \xrightarrow{\text{serialization}} \text{Serialize}(X) \xrightarrow{\text{e5-Mistral-7B-Instruct}} \Phi(X).$$

Since `e5-mistral-7b-instruct` requires input data to be formatted in the following template:

```
Instruct:     description of the classification task \n
Query:        description of the data,
```

we provide task description in the "Instruct" part, and use the serilization template to format the tabular data in the "Query" part. An illustrative example is provided in Part A-D of Figure 1, with

additional details available in Appendix A.1. Analyzing the impact of different LLM encoders, task descriptions, and serialization methods is left for future work.

## 2.2 Additional Domain Information

Another advantage of using LLM embeddings is their ability to incorporate additional domain information or prior knowledge, denoted by $C$. As demonstrated in Section 1, incorporating domain-specific information can help address $Y|X$-shifts and improve generalization performance in the target domain.

In this work, we propose a simple yet effective approach for integrating domain knowledge into tabular predictions. Rather than combining the domain information with serialized tabular data and generating a single LLM embedding, we generate separate LLM embeddings for the domain knowledge and the serialized tabular data, and then concatenate them together. The benefits of this approach are twofold: (a) although the domain information may contain significantly more words than the serialized tabular features, our concatenation method ensures a balanced 1:1 ratio between the two, preventing a single embedding that disproportionately focuses on the longer domain information; (b) by separating the tabular features from the domain information, we can efficiently update the domain information without having to regenerate all the embeddings for the entire dataset.

We explore three sources of domain information: Wikipedia, GPT-4, and labeled target samples. Given that our experiments (see Table 1 and Section 3) focus primarily on socioeconomic factors, we collected "Economy" data for each U.S. state from Wikipedia as $C$. For GPT-4, we prompt it to provide background knowledge relevant to each prediction task in each state as $C$. For labeled target samples, we serialize 32 labeled samples from the concerned domain as the prior knowledge $C$. Further details can be found at Appendix A.2. After obtaining domain information $C$, we use `e5-mistral-7b-instruct` to generate an LLM embedding for $C$. As illustrated in Parts E.2 and F.1 of Figure 1, this embedding is then concatenated with the LLM embeddings of the tabular data, which serve as input to the backend neural network models (NN). This approach allows us to generate the LLM embedding for the dataset *just once*, and subsequently concatenate it with embeddings from different prompts as needed. In Section 3, we study whether and how this additional domain information can enhance generalization under $Y|X$-shifts.

In addition, recent works on prompt engineering have focused on incorporating additional domain information to enhance prediction tasks, often through detailed instructions (Schick & Schütze, 2020; Shin et al., 2020). Our proposed framework introduces a novel approach to leveraging such information and remains fully compatible with these existing methods.

## 2.3 Model Training and Target Adaptation

**Model architecture**    For the backend model, we use a vanilla neural network (NN) classifier on both tabular features and LLM embeddings for tabular data classification. The NN is a simple feedforward neural network with several hidden layers, dropout layer, and ReLU activation functions.

When adding additional domain information via an embedding layer, the same embedding is applied to all samples from the same domain. Since the output of `e5-mistral-7b-instruct` is a 4096-dimensional vector, we simply concatenate the LLM embeddings with the embeddings of the domain information. This concatenated vector is then passed through the hidden layers, dropout layer, and ReLU activation functions. For all NNs, the final linear layer an output dimension of 2, followed by a softmax layer for binary classification. During training, we use cross-entropy as the loss function, batch size as 128, and use the Adam optimizer. Detailed model architecture and hyper-parameters are provided in the Appendix A.3 and A.4, with a discussion on hyperparameter selection provided in Section 3.1.

**Target Adaptation**    Even with the incorporation of LLM embeddings and domain information, our model may still experience $Y|X$-shifts. In practice, it is common to have a small set of samples from the target domain, which can be leveraged to better adapt the model to the target domain.

For each (source domain, target domain) pair, we begin by selecting the best training hyperparameter based on a validation criterion, which will be discussed in the Section 3.1. Using this model trained on the source domain, we explore four primary methods for target adaptation: in-context domain info, full-parameter fine-tuning, low-rank adaptation (LoRA), and prefix tuning for domain information.

Table 1: Details of datasets used in this work. "# Source→Target Pair" denotes the number of distribution shift pair for each dataset, and we consider the natural *spatial* shift between US states.

| #ID | Dataset | #Samples | #Features | Outcome | #Source Domains | #Target Domains | #Source→Target Pair |
|-----|---------|----------|-----------|---------|-----------------|-----------------|---------------------|
| 1 | ACS Income | 1.60M | 9 | Income≥50k | 51 (US States) | 50 (US States) | 2550 |
| 2 | ACS Mobility | 621K | 21 | Residential Address | 51 (US States) | 50 (US States) | 2550 |
| 3 | ACS Pub.Cov | 1.12M | 18 | Public Ins. Coverage | 51 (US States) | 50 (US States) | 2550 |

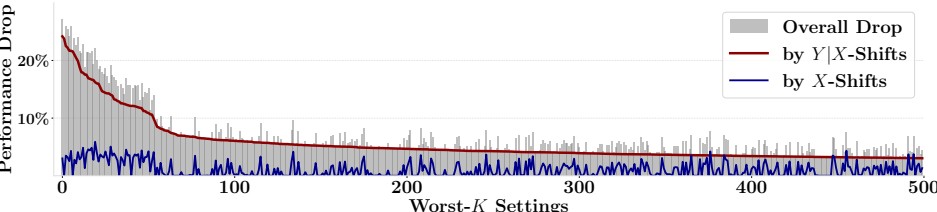

Figure 3: Shift pattern analysis. For the 2550 source→target distribution shift pairs in ACS Income dataset, we attribute the performance drop for each source→target pair into $Y|X$-shifts (red curve) and $X$-shifts (blue curve), and sort all pairs according to the drop introduced by $Y|X$-shifts. We draw the *worst-500* settings in each dataset, and the decomposition method used here is DISDE (Cai et al., 2023) with XGBoost as the reference model. Results on other datasets are in Figure 9.

For in-context domain info (F.1 of Figure 1), we keep the trained model frozen and only update the domain information, switching it from natural language description of labeled sample from the source domain during training to that of target domain during inference phase. For the other three methods, we conduct further training of the model. In full-parameter fine-tuning (F.2 of Figure 1), the entire neural network is fine-tuned using the target samples. For LoRA, we introduce a low-rank adaptation layer to each linear layer by incorporating two smaller matrices, $A$ and $B$, both with a rank of 16. Specifically, matrix $A$ has dimensions corresponding to the input size and the rank, while matrix $B$ has dimensions corresponding to the rank and the output size. Matrix $A$ is initialized with a mean of 0 and a standard deviation of 0.02, whereas matrix $B$ is initialized with zeros. These matrices are then multiplied together and added to the original weight matrix. We then fine-tune only these LoRA parameters, while keeping the rest of the model unchanged. In prefix tuning (F.3 of Figure 1), the initial domain information embedding serves as a starting point for further refinement. During training, both the the NN and the domain information embedding of the source domain are trained. For target adaptation, we switch the domain information embedding from the source to the target domain. The NN is kept frozen, and only the domain information embedding of the target domain is updated using these samples from the target domain. We refer to this process as prefix tuning.

As shown in Table 1, we use different hyperparameters for target adaptation. Detailed hyperparameters are provided in Appendix A.4, and the hyperparameter selection process is discussed in Section 3.1.

## 3 NUMERICAL EXPERIMENTS

In this section, we conduct a thorough investigation of **7650** natural shift settings (source → target domain) in 3 tabular datasets over **261,000** model configurations and summarize the observations. Our findings highlight the potential of incorporating LLM embeddings to enhance the generalization ability in tabular data prediction tasks.

### 3.1 TESTBED SETUP

**Dataset** In this work, we use the ACS dataset (Ding et al., 2021b) derived from the US-wide ACS PUMS data, where the goal is to predict various socioeconomic factors for individuals.

- ACS Income: The goal is to predict whether an individual's income is above $50K based on individual features. We filter the dataset to only include individuals above 16 years old with usual working hours of at least 1 hour per week in the past year, and an income of at least $100.

- ACS Mobility: The goal is to predict whether an individual has the same residential address as one year ago. We filter the dataset to only include individuals between the ages of 18 and 35, which increases the difficulty of the prediction task.

- `ACS Public Coverage` (abbr. as `ACS Pub.Cov`): The goal is to predict whether an individual has public health insurance. We focus on low-income individuals who are not eligible for medicare by filtering the dataset to only include individuals under the age of 65 and with an income of less than $30,000.

The details of datasets are summarized in Table 1.

**Shift Pattern Analysis**  Before benchmarking, we first analyze the shift patterns among the 2550 source→target pairs in each dataset. Specifically, we utilize DISDE (Cai et al., 2023) (reference model as XGBoost) to decompose the performance degradation from the source domain to the target into two parts: (a) $Y|X$ (concept)-shifts and (b) $X$ (covariate)-shifts. By utilizing tailored shift patterns, we can conduct an in-depth analysis of where the strength of LLM embeddings lies. As shown in Figure 3, we sort all pairs according to the strength of $Y|X$-shifts, where we find that the natural spatial shifts are mainly comprised of $Y|X$-shifts. These findings broaden the scope of the analysis in `WhyShift` (Liu et al., 2023) by examining 7,650 shift pairs, a significant increase from the 169 pairs studied in the original work.

**Algorithms**  As introduced in Section 2, we compare various methods that incorporate LLM embeddings into tabular data prediction, including different finetuning methods (no finetuning, finetuing on full parameters, and low rank adaptation (LoRA)) and different embeddings (w/ or w/o extra information). Besides, in order to fully compare the performances, we also include a wide range of learning strategies that perform on `Tabular` features, including basic models (LR, SVM, NN), tree ensembles (XGB, LGBM, GBM), robust methods (KL-DRO, $\chi^2$-DRO, Wasserstein DRO, CVaR-DRO, and Unified-DRO). All methods are summarized in Table 2.

**Experiment Setup**  We conduct experiments with more than **261,000** model configurations on 2550 source→target shift pairs in `ACS Income`, `ACS Mobility`, and `ACS Pub.Cov` datasets respectively (**7650** settings in total). For each source→target shift pairs, we randomly sample $20,000$ labeled data from the source and target domain respectively, as the training and test dataset. We evaluate the model trained on the source domain, with or without target adaptation, and report the *Macro F1 score* on the testing dataset.

Given the numerous training hyperparameters—learning rate, number of training epochs, hidden layer dimension, dropout ratio—we use a validation set of 32 randomly sampled labeled target domain samples to choose the optimal training hyperparameters, based on the highest F1 score in the validation dataset. Since our metric is Macro F1 score, the validation set is set as balanced between positive and negative classes. Note that the hyperparameter selection is *near*-oracle, as it leverages target samples, albeit in a limited quantity. When doing finetuning, we sample another 32 labeled target samples to finetune the model. And we use the same 32-sample validation dataset (for training hyperparameter selection) to select the target adaptation hyperparameters that yield the best Macro F1 score. Note that our testbed allows flexible sample sizes for training, validation, testing, and finetuning. Additionally, we perform an ablation study on different allocations of the overall target samples in validation and finetuning (see Figure 8). See details on hyperparameters in the Appendix A.4.

## 3.2 PRIMARY FINDINGS

We begin by presenting the key observations from our results. In addition to the metric equation 1 introduced in Section 1, we report performance metrics averaged over the source-target pairs.

**LLM embeddings improve performance, but when applied alone do not consistently outperform tree-ensembles.**  To better assess the generalization ability when incorporating LLM embeddings, we select the worst 500 settings (out of 2,550 total settings per dataset) based on the severity of $Y|X$-shifts and report the average Macro F1 Score in Figure 4. Each bar represents the average result across these worst 500 settings, characterized by the most severe $Y|X$-shifts. Thus, even a 1pp improvement is significant, as it implies consistent gains of about 1pp across each of the worst 500 settings.

Comparing "NN on LLM embeddings" to "NN on tabular features" (with the backbone model fixed as NN), we observe LLM embeddings significantly enhance generalization under distribution shifts on the `ACS Income` and `ACS Mobility` datasets, with average improvements of 2.4pp and 9.9pp. Notably, "NN on LLM embeddings" even outperforms XGBoost under strong distribution shifts on

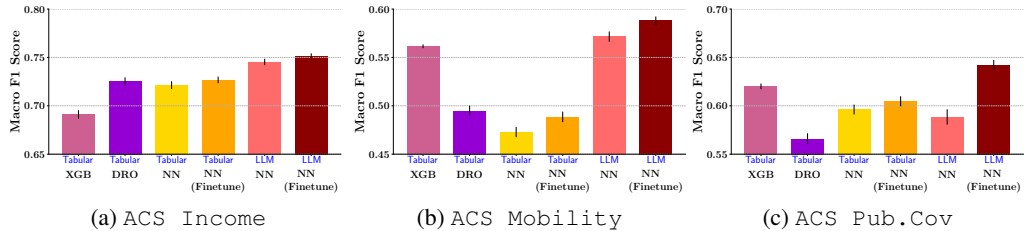

Figure 4: Average Macro F1 Score over the worst-500 settings. For each dataset, we sort the 2550 settings according to the magnitude of $Y|X$-shifts and select the **worst-500** settings. We calculate the average Macro F1 Score for each method. For all methods, we select the best hyper-parameters of the basic model according to 32 samples from the target domain. We use CVaR-DRO based on NN here to represent DRO methods. For finetuning methods, we use 32 target samples.

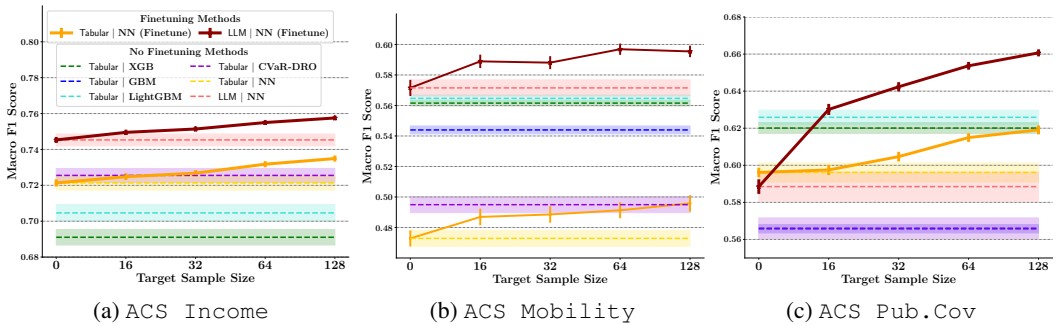

Figure 5: Average Macro F1 Score over Worst-500 settings with different #target samples. Dotted lines represent methods that do not require finetuning, while solid lines show the performance of finetuning methods relative to the number of target samples. Three figures share the same legend.

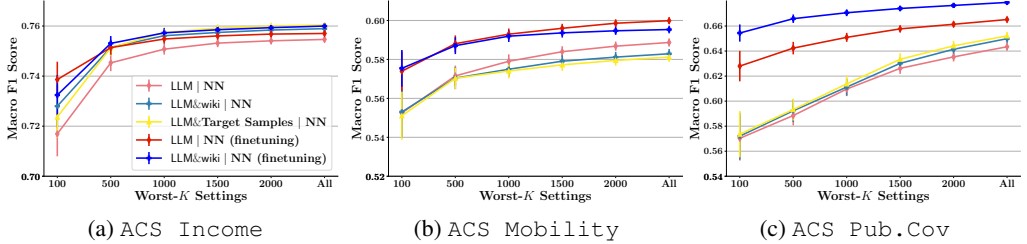

Figure 6: Average performance over the worst-$K$ pairs. For each dataset, we sort the 2550 pairs according to the magnitude of $Y|X$-shifts and select the worst-$K$ settings ($K \in \{100, 500, 1000, 1500, 2000, 2550\}$). For methods requiring finetuning, we use 32 target samples here. "LLM& Target Samples" represents the in-context domain info in F.1 of Figure 1, where we embed 32 target samples as domain information. Three figures share the same legend.

these datasets. This demonstrates the potential of LLM embeddings in tabular data prediction, where they can contribute to more generalizable models.

A different trend is observed on the `ACS Pub.Cov` dataset, where the inclusion of LLM embeddings results in a performance drop for NN models. This suggests that simply incorporating LLM embeddings does not always resolve distribution shift issues; their effectiveness may vary across datasets, particularly depending on whether the LLM embeddings provide additional relevant information for the specific prediction task.

**A small number of target samples can make a big difference.** While incorporating LLM embeddings doesn't always yield improvements, we find that even a small number of target samples can have a significant impact. As shown in 4, finetuning the "NN on LLM embeddings" model with just 32 target samples significantly improves the average performance across the worst 500 settings for both the `ACS Mobility` and `ACS Pub.Cov` datasets. Notably, for the `ACS Pub.Cov` dataset, where LLM embeddings alone provided no improvement, finetuning with only 32 target samples leads to a 5.4pp gain, even surpassing XGBoost by 2.2pp. This highlights the adaptability of LLM

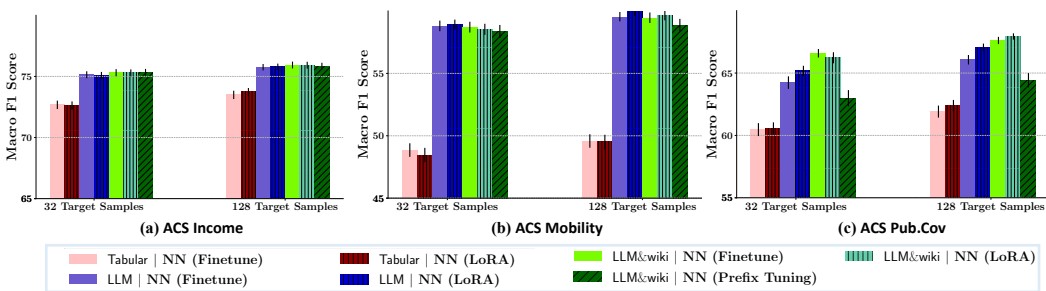

Figure 7: Comparison between full-parameter finetuning, LoRA, and prefix tuning. We show the average Macro F1 Score over the worst-500 settings.

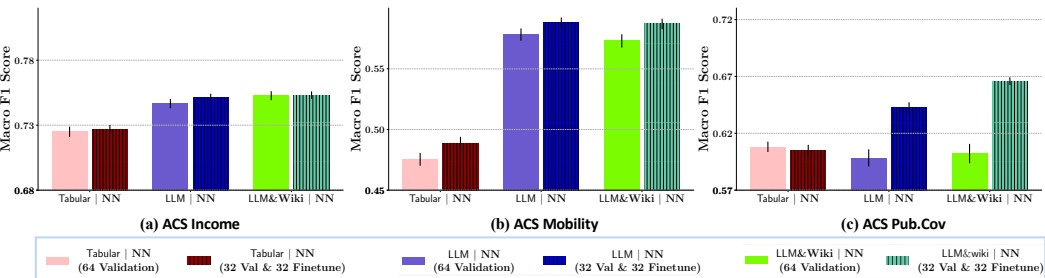

Figure 8: Comparison between different allocation of target samples into validation and finetuning. We report the average Macro F1 Score over the worst-500 settings.

embeddings, making them a promising tool for harnessing their power across various downstream real-world tasks.

Furthermore, in Figure 5, we illustrate how the performance of finetuning methods varies with different numbers of target samples. As shown, our conclusions remain consistent regardless of the number of target samples.

**The performance gain brought by target samples is larger under stronger $Y|X$-shifts.** As shown in Figure 6 (LLM | NN v.s. LLM | NN (finetuning)), for NN using LLM embeddings, finetuning with 32 target samples yields an average performance gain of 5.8pp across the worst-100 settings on ACS Pub.Cov, compared to no finetuning. This is notably higher than the 1.2pp average gain observed across all 2550 settings (**4.8 times**). Similarly, for ACS Income, it is about **9 times**.

### 3.3 AUXILIARY FINDINGS

In addition to the primary findings, we have several other noteworthy observations.

**"Right" domain information matters.** From Figure 6, we observe that additional domain information from either Wikipedia or 32 target samples (via in-context domain info, F.1 of Figure 1) does not lead to significant improvements (comparing the shallow blue and yellow curves with the shallow red curve). However, when combined with finetuning, this extra domain information performs significantly better on ACS Pub.Cov. Since finetuning can be considered a method for incorporating domain-specific information, this suggests that identifying the *right* domain information is crucial.

**Specific finetuning approaches are less crucial than expected.** Given the large number of model parameters and limited labeled target samples, one might expect parameter-efficient methods like Low-Rank Adaptation (LoRA) and Prefix Tuning to offer a clear advantage. However, as shown in 7, all methods significantly outperform the non-finetuned version when using LLM embeddings, and the choice of finetuning method appears less significant in our setting.

An exception is prefix tuning on the ACS Pub.Cov task, where performance was several percentage points lower. While this requires further investigation, our key takeaway is that under $Y|X$ shifts, 1) finetuning models using LLM embeddings can greatly enhance classification performance; 2) popular finetuning methods yield comparable results.

**Allocation of labeled target samples matters.** In Figure 8, we compare two allocation schemes of 64 labeled target samples. For the solid bars, all 64 samples are utilized as the validation dataset for hyperparameter selection. For the shaded bars, we allocate 32 samples for validation and the remaining 32 samples for finetuning. Based on this, we initially explore and understand the impact of sample allocation. For `ACS Mobility` and `ACS Pub.Cov`, target adaptation using LLM embeddings (the shaded bar) significantly outperforms the validation approach (the solid bar). However, for `ACS Income`, the improvement from target adaptation is only marginal. This shows that although target adaptation is effective, its improvement is highly dependent on the specific distribution and $Y|X$ shift level. This raises further questions about how to optimally allocate resources. Although our findings endorse considering target adaptation, identifying the best allocation strategy is left as future work.

## 4 DISCUSSION BASED ON THEORETICAL INSIGHTS

We take a brief examination of the theoretical insights that may lie behind our empirical findings. While standard generalization bounds are vacuous for neural networks, we nevertheless find that theoretical results from domain adaptation provide a useful starting point for understanding why finetuning LLM embeddings with small target samples can result in superior generalization performance under substantial $Y|X$ shifts, particularly in comparison to tabular features.

Let $\phi(X)$ denote a feature map and $Y$ a binary label. We let $P$ and $Q$ be the source and target distributions. We consider a model class $\mathcal{H}$ of VC dimension $d$. For any model $h \in \mathcal{H}$, we use $\epsilon_P(h) := \mathbb{E}_P[\mathbb{I}(h(\phi(X)) \neq Y)]$ to denote the expected 0-1 loss on the source domain and $\hat{\epsilon}_P^m(h)$ its empirical counterpart based on $m$ i.i.d. samples from $P$. We define $\epsilon_Q(h)$ and $\hat{\epsilon}_Q^m(h)$ on the target.

Although in practice we finetune on the target, we use a mixture problem as a rough approximation. Suppose that we have $(1 - \beta)m$ i.i.d. samples from source domain $P$ and $(\beta m)$ i.i.d. samples from target domain $Q$. Let $\hat{h}_{\alpha,\beta}$ be the minimizer of the $\alpha$-weighted empirical error

$$\hat{h}_{\alpha,\beta} := \arg\min_{h \in \mathcal{H}} \left\{ \alpha \hat{\epsilon}_Q^{\beta m}(h) + (1 - \alpha) \hat{\epsilon}_P^{(1-\beta)m}(h) \right\}.$$

The following classical result from Ben-David et al. (2010, Theorem 3) bounds the generalization error on the target.

**Proposition 1.** *For any $\delta \in (0, 1)$, with probability at least $1 - \delta$,*

$$
\epsilon_Q(\hat{h}_{\alpha,\beta}) - \inf_{h \in \mathcal{H}} \epsilon_Q(h) \leq 4\sqrt{\frac{\alpha^2}{\beta} + \frac{(1-\alpha)^2}{1-\beta}} \sqrt{\frac{2d \log(2(m+1)) + 2\log \frac{8}{\delta}}{m}} \\
+ 2(1-\alpha) \underbrace{d_{\mathcal{H}\Delta\mathcal{H}}(P_X, Q_X)}_{X\text{-shifts}} + 2(1-\alpha) \underbrace{\inf_{h \in \mathcal{H}} \{\epsilon_P(h) + \epsilon_Q(h)\}}_{Y|X\text{-shifts}},
\tag{2}
$$

*where $d_{\mathcal{H}\Delta\mathcal{H}}(\cdot, \cdot)$ denotes the $\mathcal{H}\Delta\mathcal{H}$-distance between two (marginal) distributions.*

Since we use limited (32) labeled target samples for finetuning, $\beta$ is small. Also, we use a smaller learning rate for finetuning than for training on the source, implying that $\alpha$ is even smaller than $\beta$. Thus, we expect the constants $\sqrt{\frac{\alpha^2}{\beta} + \frac{(1-\alpha)^2}{1-\beta}}$ and $(1-\alpha)$ to both be close to 1. Although the VC-dimension $d$ can be vacuously large for neural networks (a well-known defect in statistical learning theory), having a large enough sample size $m$ can generally make the first term comparable to the next two terms. For the raw features $\phi_{\text{tabular}}(X) = X$, we have observed empirically that $Y|X$-shifts are salient, implying that the third term (related to $Y|X$-shifts) dominates the second term (related to $X$-shifts), and it can also dominate the first term when $Y|X$ shifts are particular significant. In comparison, we conjecture that LLM embeddings $\phi_{\text{LLM}}$ can reduce the gap between $\mathbb{E}_P[\mathbb{I}(Y \neq h(z) \mid \phi_{\text{LLM}}(X) = z]$ and $\mathbb{E}_Q[\mathbb{I}(Y \neq h(z)) \mid \phi_{\text{LLM}}(X) = z]$ by *incorporating prior knowledge* encoded during LLM pre-training. Since $\epsilon_P(h) = \int \mathbb{E}_P[\mathbb{I}(Y \neq h(z)) \mid \phi(X) = z] \, dP_{\phi(X)}(z)$ (and similarly for $Q$), we thus expect the third term to be much smaller than using raw features $\phi_{\text{tabular}}(X)$. This suggests that when using LLM embeddings $\phi_{\text{LLM}}$, the generalization bound can be smaller than that of raw features $\phi_{\text{tabular}}(X)$, especially when $Y|X$ shift is more significant under raw features $\phi_{\text{tabular}}(X) = X$; recall Figure 6.

We hope our empirical findings spur future theoretical investigations into the foundations of LLM-based target adaptation.

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

## A    MODEL TRAINING DETAILS

In this section, we outline the serialization scheme, the generation of additional domain information, our model architecture, and the hyperparameters used for training and target adaptation.

### A.1    SERIALIZATION SCHEME

As discussed in Section 2.1, serializing a row of tabular data, such as $X$, requires two components: a task description and a description of the data.

**Task description**    For the ACS Income, ACS Mobility, and ACS Public Coverage datasets, we consider the same binary classification task as described in Ding et al. (2021a). We adopt concise task descriptions as suggested by Hegselmann et al. (2023), as follows:

- `ACS Income`: Classify whether US working adults' yearly income is above $50000 in 2018.
- `ACS Mobility`: Classify whether a young adult moved addresses in the last year.
- `ACS Public Coverage` : Classify whether a low-income individual, not eligible for Medicare, has coverage from public health insurance.

**Description of the data**    For all three ACS datasets, we utilize features as shown in (Ding et al., 2021a, Appendix B), along with the domain name (state) to characterize the data. Specifically, for data from the source state, we apply the source domain name, and for data from the target state, we use the target domain name.

As illustrated in Figure 1 and recommended by Hegselmann et al. (2023), we adopt a straightforward text template: "The `feature name` is `value`." However, for certain features with less common or more complex feature names, we opt for a template as "The person (appropriate verb) `value`" to more clearly convey the data description. For example, for the feature "Gave birth to child within the past 12 months" with the value "No", the serialization would be "The person did not give birth to a child within the past 12 months." The features that use this alternative template are as follows:

- `ACS Income`: class of worker;
- `ACS Mobility`: class of worker, disability, employment status of parents, citizenship, military service, hearing difficulty, vision difficulty, cognitive difficulty, grandparent living with grandchildren, employment status;
- `ACS Public Coverage`: disability, employment status of parents, citizenship, mobility, military service, hearing difficulty, vision difficulty, cognitive difficulty, gave birth to child within the past 12 months, employment status.

For features without associated values, we omit them during serialization.

### A.2    ADDITIONAL DOMAIN INFORMATION

As shown in Section 2.2, we study three sources of domain information: Wikipedia, GPT-4, and labeled target samples. As we use the `e5-mistral-7b-instruct` to generate an LLM embedding for the domain information, we need to specify both the "Instruct" and "Query" components.

For the "Instruct" part, we apply the same task description as outlined in Section A.1. For the "Query" part, we utilize various descriptions of the additional domain information:

- Wikipedia: For all three datasets, we use the "Economy" section of each state's Wikipedia page as the additional domain information.
- GPT4: For each state, we pose the following question to GPT4, and use its response as the additional domain:
  - `ACS Income`: "We aim to develop a classifier to determine whether U.S. individuals earned over $50000 in 2018, using features such as age, sex, educational attainment,

race, class of worker, marital status, occupation, and hours worked per week over the past 12 months. Given the unique economic and demographic profile of `state_name`, how might these factors influence income levels differently compared to other U.S. states? Please provide a 2000-word summary detailing these differences."

– `ACS Mobility`: "We aim to develop a classifier to determine whether a young adult moved addresses in the last year, using features such as age, sex, educational attainment, race, class of worker, marital status, occupation, total income, and hours worked per week over the past 12 months. Given the unique economic and demographic profile of `state_name`, how might these factors influence mobility levels differently compared to other U.S. states? Please provide a 2000-word summary detailing these differences."

– `ACS Public Coverage`: "We aim to develop a classifier to determine whether a low-income individual, not eligible for Medicare, has coverage from public health insurance, using features such as age, sex, educational attainment, race, disability, marital status, occupation, citizenship status, mobility status, military service, nativity, total income, and employment status. Given the unique economic and demographic profile of `state_name`, how might these factors influence public coverage levels differently compared to other U.S. states? Please provide a 2000-word summary detailing these differences."

- labeled target samples: We use the following template for 32 labeled target samples:

```
Here are some examples of the data: \n
description of one target sample \n
Answer: (Yes or No). \n \n
description of another target sample \n
Answer: (Yes or No). \n \n
...
```

We use the same data descriptions as in Section A.1, with the exception that the state/domain name is omitted, as it is represented by the labeled target samples provided.

**Shift Patterns**  The shift patterns of all settings are shown in Figure 9.

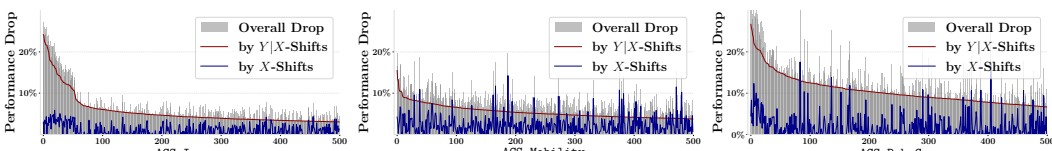

Figure 9: Shift pattern analysis. For the 2550 source→target distribution shift pairs in `ACS Income`, `ACS Pub.Cov`, `ACS Mobility` datasets respectively, we attribute the performance drop for each source→target pair into $Y|X$-shifts (red curve) and $X$-shifts (blue curve), and sort all pairs according to the drop introduced by $Y|X$-shifts. We draw the *worst-500* settings in each dataset, and the decomposition method used here is DISDE (Cai et al., 2023) with XGBoost as the reference model.

A.3  MODEL ARCHITECTURE

We detail the baselines used in our paper.

**Fully-connected Neural Networks (NN)**  Given the varying input dimensions for Tabular features, LLM embeddings, and LLM embeddings with additional domain information, we employ three neural networks with similar architectures. We then train these networks and conduct target adaptation using Empirical Risk Minimization (ERM).

For all three datasets using Tabular features, we use a hidden layer with output dimension being `hidden layer dim` as a hyperparameter, a dropuput layer with `dropout ratio` being a hyperparameter, ReLu activation. We then have another hidden layer with input and output dimension both being `hidden layer dim`, and then softmax layer with dimension 2 as the output.

For datasets using Tabular features, the network includes a hidden layer where the output dimension is set by the hyperparameter `hidden layer dim`. It is followed by a dropout layer (`dropout ratio` as a hyperparameter), ReLU activation, another hidden layer the input and output dimensions

are both equal to `hidden layer dim`. The network concludes with a softmax layer with an output dimension of 2.

For datasets using only LLM embeddings, the input dimension is 4096, which is the output dimension of `e5-mistral-7b-instruct`. The network consists of three hidden layers with fixed dimensions, where the input and output dimensions are (4096, 1024), (1024, 256), and (256, 128), respectively. Each layer uses ReLU activation. Next, there's a hidden layer with an input dimension of 128 and an output dimension set by the `hidden layer dim` hyperparameter. This is followed by a dropout layer (with `dropout ratio` as a hyperparameter), ReLU activation, and another hidden layer where both the input and output dimensions are `hidden layer dim`. A ReLU activation follows this final hidden layer, and the network concludes with a softmax layer that outputs a dimension of 2.

For datasets using LLM embeddings concatenated with additional domain information, the input dimension is 8192. We slightly update the first three hidden layers, with input and out dimensions being (8192, 2048), (2048, 512), and (512, 128), respectively. All other neural network structure and hyperparameters are the same as NNs with LLM embeddings only.

When applying low-rank adaptation (LoRA) for target adaptation, we introduce a low-rank adaptation layer to each linear layer by incorporating two smaller matrices, $A$ and $B$, both with a rank of 16. Specifically, matrix $A$ has dimensions corresponding to the input size and the rank, while matrix $B$ has dimensions corresponding to the rank and the output size. Matrix $A$ is initialized with a mean of 0 and a standard deviation of 0.02, whereas matrix $B$ is initialized with zeros. These matrices are then multiplied together and added to the original weight matrix.

**Tree Ensemble Models**     Gardner et al. (2022) show that several tree-based methods are competitive on tabular datasets. And gradient-boosted trees (e.g., XGB (Chen & Guestrin, 2016), LGBM (Ke et al., 2017), GBM (Natekin & Knoll, 2013)) are widely considered as the state-of-the-art methods on tabular data. Therefore, we compare XGB, LGBM, and GBM in this work. For GBM, we use the standard implementations in `scikit-learn` (Pedregosa et al., 2011). For XGB and LGBM, we use the standard implementations in the `xgboost` package[2] and the `lightgbm` package[3]. All these methods are trained on CPUs.

**DRO Methods**     Distributionally robust optimization (DRO) methods optimize the worst-case loss over an ambiguity set $\mathcal{P}$:

$$\min_{f \in \mathcal{F}} \sup_{Q \in \mathcal{P}} \mathbb{E}_Q[\ell(f(X), Y)]. \tag{3}$$

The ambiguity set is typically chosen as a "ball" around the training distribution $P_{\text{tr}}$

$$\mathcal{P}(d, \epsilon) = \{Q : d(Q, P_{\text{tr}}) \le \epsilon\}, \tag{4}$$

where $d(\cdot, \cdot)$ is a distance metric between probability measures and $\epsilon$ is the radius of set. When $d$ is set as the generalized $f$-divergence (including CVaR) as:

$$d(P, Q) = E_Q\left[f\left(\frac{dP}{dQ}\right)\right], \tag{5}$$

for the KL-DRO method (Hu & Hong, 2013), we use $f(x) = x \log x - (x - 1)$; for the $\chi^2$-DRO method (Duchi & Namkoong, 2019), we use $f(x) = (x - 1)^2$; for the CVaR-DRO problem (Rockafellar et al., 2000), we use $f(x) = 0$ if $x \in [\frac{1}{\alpha}, \alpha]$ and $\infty$ otherwise, and $\alpha$ controls the worst-case ratio.

For Wasserstein DRO (Blanchet et al., 2019b), we choose $d(\cdot, \cdot)$ as the Wasserstein distance. Unified-DRO (Blanchet et al., 2023b) combines Wasserstein distance and KL-divergence as $d(\cdot, \cdot)$, and we follow their initial Github codebases and hyperparameter selection when implementing these methods.

---

[2]https://pypi.org/project/xgboost/
[3]https://pypi.org/project/lightgbm/

Table 2: Summary of methodologies: As baselines, we use basic models (LR, SVM, and NN), GBDTs (XGB, LGBM, and GBM) and robust learning methods (KL-DRO, $\chi^2$-DRO, etc.). For methods utilizing LLM embeddings, we consider different ways to incorporate domain information and different model architectures and finetuning techniques.

| Name | Feature | Domain Info | Model | Adaptation | # of HPs | Part of Fig. 1 |
|------|---------|-------------|-------|------------|----------|----------------|
| LR | Tabular | - | LR | No Finetuning | 34 | - |
| SVM | Tabular | - | SVM | No Finetuning | 34 | - |
| GBDT | Tabular | - | XGB, LGBM, GBM | No Finetuning | 200 | - |
| KL-DRO | Tabular | - | SVM | No Finetuning | 138 | - |
| $\chi^2$-DRO | Tabular | - | SVM | No Finetuning | 138 | - |
| Wasserstein-DRO | Tabular | - | SVM | No Finetuning | 17 | - |
| Unified-DRO | Tabular | - | SVM | No Finetuning | 150 | - |
| CVaR-DRO | Tabular | - | NN | No Finetuning | 200 | - |
| Tabular \| NN | Tabular | - | NN | No Finetuning | 96 | - |
|  |  |  |  | Finetuning | 12 | - |
|  |  |  |  | LoRA | 15 | - |
| LLM \| NN | LLM Embeddings | - | NN | No Finetuning | 96 | E.1 |
|  |  |  |  | Finetuning | 12 | F.2 |
|  |  |  |  | LoRA | 15 | F.2 |
| LLM & Wiki/GPT4 \| NN | LLM Embeddings | Wikipedia or GPT4 | NN | No Finetuning | 48 | E.2 |
|  |  |  |  | Finetuning | 12 | F.2 |
|  |  |  |  | LoRA | 15 | F.2 |
|  |  |  |  | Prefix Tuning | 18 | F.3 |
| LLM & In-Context Domain Info \| NN | LLM Embeddings | Labeled Target Samples | NN | No Finetuning | 96 | F.1 |

## A.4 HYPERPARAMETERS FOR TRAINING AND TARGET ADAPTATION

For each algorithm, we maintain a grid of candidate hyperparameters as shown in Tables 3 and 4. and perform hyperparameter selection as described in Section 3.1. When the number of hyperparameter configurations exceeds 200, we randomly select 200 configurations to reduce computational cost and maintain fairness in the comparison across all algorithms.

Table 3: Training hyperparameter grids used in all experiments. $\diamond$ : for methods with the total grid size above 200, we randomly sample 200 configurations for fair comparisons.

| Model | Feature | Domain Info | # of HPs | Hyperparameter | Value Range |
|---|---|---|---|---|---|
| SVM | Tabular | - | 96 | C
Kernel
Loss
$\gamma$ | $\{1e^{-2}, 1e^{-1}, 1, 1e^1, 1e^2, 1e^3\}$
$\{linear, RBF\}$
Squared Hinge
$\{0.1, 0.3, 0.5, 1, 1.5, 2, scale, auto\}$ |
| LR | Tabular | - | 23 | $L_2$ penalty | $\{1e^{-3}, 3e^{-3}, 5e^{-3}, 7e^{-3}, 1e^{-2},$
$3e^{-2}, 5e^{-2}, \dots, 1.3, 1.7, 5\}$
$1e^1, 5e^1, 1e^2, 5e^2, 1e^3, 5e^3, 1e^4\}$ |
| Tabular \| NN | Tabular | - | 96 | Learning Rate
Hidden Layer Dim
Dropout Ratio
Train Epoch | $\{0.001, 0.003, 0.005, 0.01\}$
$\{16, 32, 64, 128\}$
$\{0, 0.1\}$
$\{50, 100, 200\}$ |
| GBM | Tabular | - | 1680$^\diamond$ | Learning Rate
Num. Estimators
Max Depth
Min. Child Samples | $\{1e^{-2}, 1e^{-1}, 5e^{-1}, 1\}$
$\{32, 64, 128, 256\}$
$\{2, 4, 8, 16\}$
$\{1, 2, 4, 8\}$ |
| LGBM | Tabular | - | 1680$^\diamond$ | Learning Rate
Num. Estimators
$L_2$-reg.
Min. Child Samples
Column Subsample Ratio (tree) | $\{1e^{-2}, 1e^{-1}, 5e^{-1}, 1\}$
$\{64, 128, 256, 512\}$
$\{0, 1e^{-3}, 1e^{-2}, 1e^{-1}, 1\}$
$\{1, 2, 4, 8, 16, 32, 64\}$
$\{0.5, 0.8, 1.\}$ |
| XGB | Tabular | - | 1944$^\diamond$ | Learning Rate
Min. Split Loss
Max. Depth
Column Subsample Ratio (tree)
Column Subsample Ratio (level)
Max. Bins
Growth Policy | $\{0.1, 0.3, 1.0, 2.0\}$
$\{0, 0.1, 0.5\}$
$\{4, 6, 8\}$
$\{0.7, 0.9, 1\}$
$\{0.7, 0.9, 1\}$
$\{128, 256, 512\}$
$\{Depthwise, Loss Guide\}$ |
| KL-DRO | Tabular | - | 117 | Uncertainty Set Size $\epsilon$ | $\{1e^{-4}, \dots, 0.01, \dots, 0.99\}$ |
| $\chi^2$-DRO | Tabular | - | 117 | Uncertainty Set Size $\epsilon$ | $\{1e^{-4}, \dots, 0.01, \dots, 0.99\}$ |
| Wasserstein-DRO | Tabular | - | 138 | Uncertainty Set Size $\epsilon$ | $\{1e^{-4}, \dots, 0.01, \dots, 0.99, \dots, 3\}$ |
| CVaR-DRO | Tabular | - | 1620$^\diamond$ | Worst-case Ratio $\alpha$
Underlying Model Class | $\{0.01, 0.1, 0.2, 0.3, 0.5, 1.0\}$
NN |
| Unified-DRO | Tabular | - | 180 | Distance Type
Uncertainty Set Size $\epsilon$
$\theta_1$ | $L_{\inf}$
$\{1e^{-3}, \dots, 9e^{-1}\}$
$\{1.001, 1.01, 1.1, 1.5, 2, 3, 5, 10, 50, 100\}$ |
| LLM \| NN | LLM | - | 48 | Learning Rate
Hidden Layer Dim
Dropout Ratio
Train Epoch | $\{0.001, 0.01\}$
$\{32, 64, 128\}$
$\{0, 0.1\}$
$\{100, 200, 300, 500\}$ |
| LLM & In-Context Domain Info/Wiki/GPT4 \| NN | LLM | Labeled Target Samples or Wikipedia or GPT4 | 48 | Learning Rate
Hidden Layer Dim
Dropout Ratio
Train Epoch | $\{0.001, 0.01\}$
$\{32, 64, 128\}$
$\{0, 0.1\}$
$\{100, 200, 300, 500\}$ |

Table 4: Target adaptation hyperparameter grids used in all experiments.

| Model | Target Adaptation Method | # of HPs | Hyperparameter | Value Range |
|---|---|---|---|---|
| Tabular \| NN or LLM \| NN or LLM & Wiki/GPT4 \| NN | Finetuning | - | 12 | Learning Rate
Target adaptation Epoch | $\{10^{-7}, 10^{-6}, 10^{-5}, 10^{-4}\}$
$\{25, 50, 100\}$ |
| Tabular \| NN or LLM \| NN or LLM & Wiki/GPT4 \| NN | LoRA | - | 12 | Learning Rate
Target adaptation Epoch | $\{10^{-6}, 10^{-5}, 10^{-4}, 10^{-3}, 0.01\}$
$\{25, 50, 100\}$ |
| LLM & Wiki/GPT4 \| NN | Prefix Tuning | - | 18 | Learning Rate
Target adaptation Epoch | $\{10^{-5}, 10^{-4}, 10^{-3}, 0.01, 0.05, 0.1\}$
$\{25, 50, 100\}$ |

