# OpenReview forum: "LLM Embeddings Improve Test-Time Adaptation to Tabular $Y|X$-Shifts"
_ICLR.cc/2025/Conference — Submitted to ICLR 2025_

### Official Review · Reviewer_8GFo · 2024-10-28

**Soundness:** 2
**Presentation:** 3
**Contribution:** 1
**Rating:** 3
**Confidence:** 5

**Summary:**

This paper focuses on the test-time adaptation for tabular data with Y|X shifts. Specifically, the proposal converts the raw tabular data into a new embedding with LLM and trains a shallow neural network on the embedding. When facing the distribution shift, the authors propose to fine-tune the network for the adaptation. Experimental results on three datasets are reported.

**Strengths:**

1) The paper focuses on the test-time adaptation for tabular data with Y|X shift. The problem is important and has been rarely studied.
2) The proposal is clear and expected to be effective.

**Weaknesses:**

1) The technical contribution is limited. The paper is more like an engineering work. The adopted techniques are all widely-adopted techniques for tabular data learning and the conclusions are also not surprising. It is not clear what technical problems have been addressed or what new insights have been proposed by this paper.
2) For the experimental results, the authors only adopted three datasets and the feature dimensions are small-scale.
3) How many times about the experiments repeated and how about the performance variance?
4) The authors claim that the LLM embeddings improve performance. However, the LLM embeddings are influenced by the serialization method and the choices of different LLMs. The influence of these should be discussed.
5) It is impractical to know the distribution shift in advance. In more practical scenarios, we need to detect the distribution shift first and then adapt.

**Questions:**

As discussed above.

---

> ### Author Response · Authors · 2024-11-21
>
> ### **Q1. Contribution and Novelty**
> Please refer to [Contribution and Novelty] in the `General Response 1`.
>
> ### **Q2. Dataset diversity**
> Please refer to [Dataset diversity] in the `General Response 2`.
>
> ### **Q3. Influence of LLM and serialization method**
> Our main contribution is to show that for the challenging and underexplored problem of tabular classification under $Y|X$-shifts, LLM embeddings can significantly improve model performance, even using standard and recent serialization method from [1] and LLM encoder from [2]. As we stated in Lines 053 and 200, exploring alternative methods is left for future exploration.
>
> ### **Q4. Feature dimension**
> We respectfully disagree. Our feature dimensions are standard for tabular classification: ACS Income has 9 features, ACS Mobility 21, and ACS Public Coverage 18. Similarly, the well-known TabLLM [1], which explores LLMs for tabular classification, examines 9 datasets with feature dimensions ranging from 4 to 20.
>
> ### **Q5. [Variance of experiments**
> We've reported the experimental variance by presenting F1 scores across different $Y|X$-shifts, with standard errors shown as error bars in Figures 4–8.
>
> ### **Q6. Unknown Y|X shift**
> We respectfully disagree. The inability to detect distribution shifts is precisely the core challenge of our setting and the broader distribution shift literature, as highlighted in works like the well-known GroupDRO paper. Our contribution demonstrates that, even without detecting distribution shifts, significant performance improvements can still be achieved using LLM embeddings.
>
> [1] Hegselmann, Stefan, et al. "Tabllm: Few-shot classification of tabular data with large language models." International Conference on Artificial Intelligence and Statistics. PMLR, 2023.
>
> [2] Wang, Liang, et al. "Improving text embeddings with large language models." arXiv preprint arXiv:2401.00368 (2023).
>
> [3] Sagawa, Shiori, et al. "Distributionally robust neural networks for group shifts: On the importance of regularization for worst-case generalization." arXiv preprint arXiv:1911.08731 (2019).

---

> > ### Author Response · Authors · 2024-11-23
> >
> > Dear Reviewer 8GFo,
> >
> > We are sincerely grateful for your time and efforts in the review process. Thank you so much for your support!
> >
> > We hope that our responses have been helpful. Given the limited time left in the discussion period, please let us know if there are any leftover concerns and if there is anything else we could do to address any further questions or comments. We are looking forward to your further feedback!
> >
> > Thank you!
> >
> > Paper Authors

---

> ### Comment · Reviewer_8GFo · 2024-11-26
> **Thanks for the response**
>
> Thanks for the author's response. However, after reading other reviewers' comments and the author's response. My concerns about the paper are not addressed. Firstly,  the contribution is over-claimed, since no insightful techniques for the tabular shift problem have been proposed. Secondly, many related important works have not been compared. Therefore, I do not believe the author can address these problems in the rebuttal phase. For now, the quality of this paper is far below the standard of ICLR.

---

> > ### Author Response · Authors · 2024-11-26
> > **Response to 8GFo**
> >
> > Thank you for your feedback. Below is our detailed response:
> >
> > **(1) Our novelty:**
> >
> > We respectfully disagree with the concern regarding novelty.
> >
> > Our key contribution lies in **connecting LLMs with the important yet under-explored domain of tabular Y|X shifts** [1] and demonstrating that LLMs deliver significant performance improvements. While we do not introduce new techniques, by bridging LLMs and tabular Y|X shifts, we **open up a promising research direction in this critical and under-explored area**.
> >
> >
> > **(2) Comparison to other works:**
> >
> > We respectfully disagree with the concern regarding baseline coverage.
> >
> > We have already included a broad spectrum of methods, where we
> > - cover as many types of methods as possible, including basic models, deep models (TabPFN), robust methods (DROs), GBDTs, and in-context learning;
> > - select the SOTA methods (**TabPFN and GBDTs**) according to previous benchmark papers [1,2]. Specifically, [2] compared 19 algorithms across 176 datasets, and found TabPFN outperforms other NN methods like FTTransformer, ResNet, TabNet, etc.
> >
> > We summarize the results of different methods in Figure 2, which shows that our proposed LLM embedding-based models significantly **outperform all these SOTA methods**, including **TabPFN, GBDTs**, and in-context learning by **GPT-4o-mini**.
> >
> > ****
> > [1] Liu, Jiashuo, et al. "On the need for a language describing distribution shifts: Illustrations on tabular datasets." Advances in Neural Information Processing Systems 36 (2023).
> >
> > [2] McElfresh, et al. “When Do Neural Nets Outperform Boosted Trees on Tabular Data?”. NeurIPS 2023.

---

### Official Review · Reviewer_MKos · 2024-11-02

**Soundness:** 3
**Presentation:** 3
**Contribution:** 2
**Rating:** 5
**Confidence:** 2

**Summary:**

This paper studies distribution shifts in tabular data by using embeddings from large language models (LLMs) to enhance test-time adaptation. The paper uses a LLM encoder  to featurize tabular data and fit a shallow neural network on these embeddings for tabular prediction. The paper investigates different configurations including various training algorithms, incorporating additional domain information or not, fine-tuning the shallow neural network with few target domain samples or not. The paper provides comprehensive experiments on these configurations.

**Strengths:**

1.  The authors provide a broad empirical evaluation across thousands of source-target pairs, diverse model configurations, and multiple algorithms, which strengthens the credibility of their findings. The paper includes a variety of performance metrics to assess the effectiveness of LLM embeddings under different conditions.

2. By highlighting different scenarios and distributional changes, the authors clarify the conditions under which the approach excels or may face limitations. This transparency in results provides a balanced view of the method's robustness and allows researchers to see where it might or might not be effective.

3. The authors include a focused analysis on the effectiveness of few-shot adaptation, showing how minimal labeled data in the target domain can significantly improve performance.

**Weaknesses:**

1. The approach essentially applies existing LLMs as embeddings for tabular data without introducing substantial new methods for embedding generation or adaptation. Serializing tabular data into texts and using LLMs to extract features/make predictions have been studied in previous works. The adaptation technique described (using few-shot labeled samples for fine-tuning) is a well-known concept in transfer learning and domain adaptation. Few-shot learning techniques have been applied across various fields, and using minimal labeled data to adapt models to new domains is not unique to this work.

2. Although the paper provide comprehensive experiments on the configurations of the proposed method, it is still unclear to position how the proposed method performs in the literature. Specifically, the paper compares with baselines of Tabular Data + NN, where NN is the fully-connected neural networks. However, there are rich literatures on more advanced neural networks tailored for tabular data (e.g., FT-Transformer and many others). In addition, LLMs have been used for tabular data recently [1]. Comparisons with state-of-the-art methods on neural networks for tabular data and LLMs for tabular data are expected. Having said this, I'm not asking the authors to add more experiments in the rebuttal stage.

[1] Fang, Xi, Weijie Xu, Fiona Anting Tan, Jiani Zhang, Ziqing Hu, Yanjun Jane Qi, Scott Nickleach, Diego Socolinsky, Srinivasan Sengamedu, and Christos Faloutsos. "Large language models (LLMs) on tabular data: Prediction, generation, and understanding-a survey." (2024).

**Questions:**

I suggest the authors to provide a conclusion of the paper summarising the key findings and discussing the limitations.

---

> ### Author Response · Authors · 2024-11-21
>
> ### **Q1. Contribution and Novelty**
> Please refer to [Contribution and Novelty] in the `General Response 1`.
>
> ### **Q2. Comparison to SOTA methods**
>
> As for baselines, we
> - cover as many types of methods as possible, including basic models, deep models (TabPFN), robust methods (DROs), GBDTs, and in-context learning;
> - select the ***SOTA methods (TabPFN and GBDTs)***  according to previous benchmark papers [1,2]. Specifically, [2] compared 19 algorithms across 176 datasets, and found TabPFN outperforms other NN methods like FTTransformer, ResNet, TabNet, etc.
>
> We summarize the results of different methods in Figure 2, which shows that our proposed LLM embedding-based models significantly outperform all these SOTA methods, including TabPFN, GBDTs, and in-context learning by GPT-4o-mini.
>
>
> [1]  Liu, Jiashuo, et al. "On the need for a language describing distribution shifts: Illustrations on tabular datasets." Advances in Neural Information Processing Systems 36 (2023).
>
> [2] McElfresh, et al. “When Do Neural Nets Outperform Boosted Trees on Tabular Data?”. NeurIPS 2023.
>
>
>
> ### **Q3. Conclusion**
> Thanks for your suggestion. Will add in the final version.

---

> > ### Author Response · Authors · 2024-11-23
> >
> > Dear Reviewer MKos,
> >
> > We are sincerely grateful for your time and efforts in the review process. Thank you so much for your support!
> >
> > We hope that our responses have been helpful. Given the limited time left in the discussion period, please let us know if there are any leftover concerns and if there is anything else we could do to address any further questions or comments. We are looking forward to your further feedback!
> >
> > Thank you!
> >
> > Paper Authors

---

### Official Review · Reviewer_ZwTt · 2024-11-04

**Soundness:** 2
**Presentation:** 3
**Contribution:** 2
**Rating:** 3
**Confidence:** 4

**Summary:**

This paper tackles an exciting problem by utilizing LLM embeddings for tabular classification. The authors conduct comprehensive experiments to assess the effectiveness of LLM embeddings in tabular tasks, with promising results that highlight a new and potentially impactful direction for advancing tabular data analysis.

**Strengths:**

1. The paper is well-structured and easy to follow, with the three technical components clearly and accessibly presented.
2. The experiments are comprehensive, which can demonstrate the claims of this paper.

**Weaknesses:**

1. The dataset variety is limited, as experiments are conducted on only three datasets within the same domain, which restricts the generalizability of the results and may affect the robustness of the findings.
2. The proposed method incorporates additional domain knowledge to enhance classification performance. However, it is unclear whether comparisons with tree-based methods are entirely fair, given that such methods may not effectively leverage domain knowledge. A thorough discussion on the source of performance improvements: whether from domain knowledge integration or specific technical innovations—would add valuable clarity.
3. The theoretical analysis may not be entirely appropriate for this context. Although the paper introduces a few-shot prompt tuning approach to maximize the use of limited labeled data, the theorem presented assumes a large  $m$  to achieve a small generalization bound, which contradicts the few-shot setting.
4. The term “test-time adaptation” in the title may be misleading, as it typically refers to model tuning using only unlabeled test data, which does not align with the approach described in this paper.

**Questions:**

Please refer to the "Weaknesses" section and answer the following questions.
1. Why does this paper lack a conclusion section?
2. How are prompt templates tailored for different datasets? Typically, datasets with varying columns and feature types require significantly different prompts to optimize the effectiveness of the LLM.

---

> ### Author Response · Authors · 2024-11-21
>
> We would like to address your concerns as follows, and all discussion will be incorporated into the final version. Please feel free to ask anything that remains unclear to you.
>
> ### **Q1. Dataset diversity**:
> Please refer to [Dataset diversity] in the `General Response 1`.
>
>
> ### **Q2. Additional domain info**:
> We respectfully disagree. Our method have significant improvements even without additional domain information, as shown in LLM | NN (Finetuning) in Figure 4. We also discuss the impact of additional domain information (e.g., from Wikipedia) through ablation studies in Lines 471–477 and Figure 6.
>
> ### **Q3. $m$ in our theory**:
> We respectfully disagree. As stated in Line 508-510, $m$ refers to the ***total number of samples*** from both the source and target domains, which can be large, as our source domain alone contains over 20K samples.
>
> ### **Q4. Conclusion section**
> Thanks for your suggestion. Will add in the next version.
>
>
> ### **Q5. Prompt template**
> We use the standard template for all classification tasks, as suggested by [1]. The template is like:
>
> ```
> Instruct: Description of the classification task\n
> Query: Description of the data
> ```
> which is shown in line 212~214. More types of templates can be incorporated in the future, but beyond the current scope.
> And each data point is serialized as: "The {feature name} is {feature value}."
>
> [1] Wang, Liang, et al. "Improving text embeddings with large language models." arXiv preprint arXiv:2401.00368 (2023).

---

> > ### Author Response · Authors · 2024-11-23
> >
> > Dear Reviewer ZwTt,
> >
> > We are sincerely grateful for your time and efforts in the review process. Thank you so much for your support!
> >
> > We hope that our responses have been helpful. Given the limited time left in the discussion period, please let us know if there are any leftover concerns and if there is anything else we could do to address any further questions or comments. We are looking forward to your further feedback!
> >
> > Thank you!
> >
> > Paper Authors

---

### Official Review · Reviewer_Tko8 · 2024-11-04

**Soundness:** 3
**Presentation:** 3
**Contribution:** 3
**Rating:** 6
**Confidence:** 3

**Summary:**

This paper proposes a method to solve the Y|X shift problem in the tabular data domain using LLM embeddings. Under the LLM+NN model architecture, it uses concatenation to combine the domain information of LLM with the original tabular data information for improving prediction robustness and performance. The paper study models that are easy to adapt to the target domain even with few labeled examples so that fine-tuning the classification model with a small number of samples can further improve performance. Experiments can prove the effectiveness of the proposed method.

**Strengths:**

1. The motivation is clear and the algorithm is sensible.
2. The proposed method is tested on several benchmarks.
3. The proposed method is easy to understand and simple. It does not require a large number of LLM calls and the computational complexity is not high.

**Weaknesses:**

1. On adaptation stage: The method requires the target's true label to guide the model parameter update. This setting is different from the basic test-time adaptation (TTA) setting, which is generally an unsupervised objective function [1]. The setting in this article is more like a fine-tuning setting.
2. The experiments in this article are mainly based on the ACS dataset. In recent studies, there are some other benchmarks like TableShift [2] contains Y|X shifts datasets. In the report they provided, ACS Income and ACS Pub.Cov dataset seem not have severe Y|X shifts. In order to prove the robustness to severe shifts, I think experiments on those datasets like ACS Unemployment and ICU Length of Stay are necessary.

[1] Wang, Dequan, et al. "Tent: Fully test-time adaptation by entropy minimization." arXiv preprint arXiv:2006.10726 (2020).

[2] Gardner, Josh, Zoran Popovic, and Ludwig Schmidt. "Benchmarking distribution shift in tabular data with tableshift." Advances in Neural Information Processing Systems 36 (2024).

**Questions:**

1. Figure 3: Recent study shows that label shift in tabular data causes performance degrade [1], why not take label shift into account?
2. Are Y|X shifts and concept shift the same in meaning?
3. How does proposed compare to state-of-the-art methods designed for tabular TTA [2]?
4. How to evaluate Y|X shifts degree?

[1] Gardner, Josh, Zoran Popovic, and Ludwig Schmidt. "Benchmarking distribution shift in tabular data with tableshift." Advances in Neural Information Processing Systems 36 (2024).

[2] Ren, Weijieying, et al. "TabLog: Test-Time Adaptation for Tabular Data Using Logic Rules." Forty-first International Conference on Machine Learning.

---

> ### Author Response · Authors · 2024-11-21
>
> Thank you very much for your support. We would like to address your concerns as follows:
>
> ### **Q1. Dataset Diversity**:
>
> Please refer to `General Response 1`.
>
>
> ### **Q2. Label and concept shift**:
> In [1], the authors classify distribution shifts into three categories: covariate shifts (changes in $P(x)$), label shifts (changes in $P(y)$), and concept shifts (changes in $P(y∣x)$).
>
> We use a different way and categorize distribution shifts into a combination of $X$-shifts (equivalent to covariate shifts) and $Y∣X$-shifts (equivalent to concept shifts). This is based on the observation that $p(y) = p(x)⋅p(y∣x)$, and we can changes in $P(y)$ is equivalent to a combination of covariate shift and concept shifts. (See further discussion in `Q3: Evaluation of Y∣X-Shift Degree`.)  We will add comparison to [1] in the next version.
>
> ### **Q3 . Evaluation of $Y|X$-shift degree**
>
>
> We use DISDE [2] to attribute the performance drop of XGB between the source and target domains to two factors: X-shifts and Y|X-shifts. The results, presented in Figure 3 and Figure 9 (Appendix), reveal that the majority of the performance decline is driven by Y|X-shifts. Additionally, the degree of Y|X-shift varies across different (source, target) pairs.
>
> ### **Q4. Severe Y|X shifts in ACSIncome, ACS Pub. Cov**
>
> We believe Y|X shifts are severe in ACS Income and ACS Pub. Coverage due to two reasons:
> 1. as shown in Figure 3 and Figure 9 in Appendix, for most (source, target) pair,  the majority of performance drops are attributed to Y|X-shifts rather than X-shifts. This is  different from findings of [1], possibly because [1] include an additional category, yet we further consider how label shifts can be attributed to X and Y|X shifts.
> 2. as reported in [3], a recent benchmarking paper for tabular Y|X shifts, they report ACS Income to have the most severe Y|X shifts among 7 types of datasets they consider.
>
> That said, we acknowledge there may exist additional datasets with more severe Y|X shifts. We have further included the USAccident dataset (see [Dataset Diversity] in the `General Response 2`) and are open to incorporating more datasets based on your suggestions.
>
> ### **Q5.  Comparison to tabular TTA [4]**
> Our approach differs from [4] primarily in our assumptions about distribution shifts. While [4] assumes that the logical structure of the rules remains invariant despite shifts between the source and target domains, we make no such assumptions. This invariance allows [4] to achieve strong performance on the target domain without requiring labeled target samples. However, as we discuss in Lines 035–039, under general Y|X shifts, the source data may offer limited insight into the target distribution. This makes it infeasible to generalize to a completely new and unknown domain without any labeled information from the target domain and we must assume access to labeled target samples.  We will clarify these distinctions in the next version.
>
> [1] Gardner, Josh, Zoran Popovic, and Ludwig Schmidt. "Benchmarking distribution shift in tabular data with tableshift." Advances in Neural Information Processing Systems 36 (2024).
>
> [2] Cai, Tiffany Tianhui, Hongseok Namkoong, and Steve Yadlowsky. "Diagnosing model performance under distribution shift." arXiv preprint arXiv:2303.02011 (2023).
>
> [3]  Liu, Jiashuo, et al. "On the need for a language describing distribution shifts: Illustrations on tabular datasets." Advances in Neural Information Processing Systems 36 (2023).
>
> [4] Ren, Weijieying, et al. "TabLog: Test-Time Adaptation for Tabular Data Using Logic Rules." Forty-first International Conference on Machine Learning.

---

> > ### Author Response · Authors · 2024-11-23
> >
> > Dear Reviewer Tko8,
> >
> > We are sincerely grateful for your time and efforts in the review process. Thank you so much for your support!
> >
> > We hope that our responses have been helpful. Given the limited time left in the discussion period, please let us know if there are any leftover concerns and if there is anything else we could do to address any further questions or comments. We are looking forward to your further feedback!
> >
> > Thank you!
> >
> > Paper Authors

---

> ### Comment · Reviewer_Tko8 · 2024-11-26
>
> Thanks for the author's response. My concern about W1 still remain. The setting is more like fine-tuning setting instead of test-time adaptation, which are not addressed in your response and may cause misleading. Additionally, the result may be reasonable when considering x-shifts, y-shifts and Y|X-shifts together. I hope the author can apply the analysis and make this paper better.

---

> ### Author Response · Authors · 2024-11-26
> **Response to Tko8's concern on (1) paper title; (2) diversity of distributuon shifts**
>
> Thanks for your thoughtful feedback. We greatly appreciate your efforts in helping us improve the paper.
>
> **(1) Paper Title:**
>
> We will change the title to "LLM improves tabular Y|X shifts through few-shot fine-tuning", or other similar variations.
>
> While TTA (Test-Time Adaptation) does not exclude using labeled target samples by its name, we fully understand your concern that TTA is most commonly associated with use of unlabeled target samples, and the current title can potentially lead to confusion among readers. The title change will not affect our findings, but we hope it will clarify our scope and address your concerns.
>
> **(2) Diversity of Distributional Shifts:**
>
> We believe our paper already considers X-shifts, Y-shifts, Y|X-shifts together.
>
> - As discussed in the `General response #2`, since p(y) = p(x) * p(y|x), **Y-shifts can be decomposed into a combination of X-shifts and Y|X-shifts**.
>
> - Our current datasets already contains 7650 (source, target) pairs that corresponds to **different combination of X-shifts and Y|X shifts**.
>
>     - This is supported by Figure 3 and Figure 9 (Appendix), where we use DISDE [1] to show different (source, target) pairs have different levels of X-shift and Y|X shifts.
>
>     - Also, our paper is a generalization of the SOTA benchmark paper for tabular distributional shifts [2], and considers more (source target) pairs.
>
>
>
> - We have **added an additional dataset USAccident** to refelct real-life and more diverse distributional shifts. LLM embeddings still yields huge improvement than tabular embeddings (See `General response #1`)
>
> - **We do not focus solely on Y|X shifts.** Rather, we use tabular Y|X shifts in the title to emphasize that we allow Y|X shifts, which sets our work apart from many prior studies that primarily concentrate on X-shifts.
>
> - We will **add comparison to [3]** to further explain how our categorization of distribution shifts is related to label shifts, concept shifts, and covariate shifts introduced in [3].
>
>
> We hope these clarifications address your concerns, and we welcome further discussion to refine the paper further. Thank you again for your valuable feedback.
>
>
> ****
>
> [1] Cai, Tiffany Tianhui, Hongseok Namkoong, and Steve Yadlowsky. "Diagnosing model performance under distribution shift." arXiv preprint arXiv:2303.02011 (2023).
>
> [2] Liu, Jiashuo, et al. "On the need for a language describing distribution shifts: Illustrations on tabular datasets." Advances in Neural Information Processing Systems 36 (2023).
>
> [3] Gardner, Josh, Zoran Popovic, and Ludwig Schmidt. "Benchmarking distribution shift in tabular data with tableshift." Advances in Neural Information Processing Systems 36 (2024).

---

### Official Review · Reviewer_pBwo · 2024-11-08

**Soundness:** 3
**Presentation:** 3
**Contribution:** 3
**Rating:** 6
**Confidence:** 4

**Summary:**

This paper investigates the challenges posed by shifts in tabular data, specifically shifts in the relationship between labels and covariates (Y|X-shifts). These shifts often occur due to missing variables or confounders, making it difficult for models trained on a source domain to generalize to a target domain.

The authors propose a solution using embeddings generated from large language models (LLMs) to encode tabular data. They demonstrate that LLM embeddings can improve adaptation to target domains even with limited labeled target samples, as the embeddings incorporate a broader contextual knowledge that helps mitigate the effects of these Y|X-shifts. Their study includes a comprehensive evaluation across 7,650 source-target pairs and comparisons with 20 algorithms, revealing that shallow neural networks (NNs) trained on LLM embeddings outperform other models in many cases.

The study's results highlight that while LLM embeddings alone do not consistently surpass state-of-the-art tree-based models, their performance significantly improves when adapted with a small number of target samples. Specifically, using 32 labeled target samples for fine-tuning yields substantial gains across multiple datasets, especially under scenarios of strong Y|X-shifts. The authors also explore different methods of adaptation, including in-context learning, low-rank adaptation, and prefix tuning, finding that fine-tuning with LLM embeddings provides a practical approach for generalizing across tabular Y|X-shifts.

**Strengths:**

This paper excels in addressing a pressing issue in machine learning: adapting models to distributional shifts in tabular data, specifically Y|X-shifts. The authors take an innovative approach by leveraging LLM embeddings to encode tabular data, which allows the model to incorporate broader contextual knowledge. This strength is significant because it enables the model to adapt to changes in label-covariate relationships, even with minimal (32 examples) labeled data from the target domain. The comprehensive evaluation of 7,650 source-target pairs across three datasets (ACS Income, Mobility, and Public Coverage) highlights the robustness of their approach. By testing against 20 algorithms (traditional GBDT and NN based models) and exploring a vast configuration space, the authors provide a solid foundation for the efficacy of LLM embeddings in generalizing across diverse tabular data environments. This thorough evaluation not only strengthens the empirical findings but also sets a new benchmark in evaluating distributional shifts in tabular datasets.

**Weaknesses:**

This method is only applicable when there is a description available for all features of the tabular data. It would be good to expand this method to datasets which include features without any description. A possible solution can be by learning embeddings of such features from scratch and utilizing them as is done in TabTransformer. This will provide a comprehensive solution for different types of tabular data.

**Questions:**

The proposed method appears limited to datasets where all features have available descriptions. Expanding this approach to include datasets with features lacking descriptions would increase its versatility. One possible solution could involve learning embeddings for such features from scratch, similar to methods like TabTransformer, which can effectively handle mixed data types by embedding categorical features without requiring explicit descriptions.

It would be beneficial for the authors to discuss this limitation in the manuscript and potentially compare their method to approaches like TabTransformer that support feature embedding without descriptions. This addition would provide a clearer understanding of the method’s scope and suggest pathways for future improvements in handling various types of tabular data.

---

> ### Author Response · Authors · 2024-11-21
>
> Thank you very much for your positive review. We would like to address your concerns as follows:
>
> ### **Q1. Datasets lacking feature descriptions**
>
> Thank you for your suggestions. Feature descriptions can be categorized into two types: (1) feature names, such as "state" or "sex" for each column, and (2) categorical descriptions, which explain the meaning of each categorical value, e.g., 1 for CA and 2 for NY.
> - **Feature descriptions are essential for LLMs in tabular classification**, as the R1 (pBwo) noted. Even if we have complete categorical descriptions, the feature names alone play an essential role. TabLLM [1] shows that removing feature names in serialization results in a performance drop in their Figure 2 (Text Template vs. List Only Values).
> - **Our main focus in robustness under $Y|X$-shifts**. Feature descriptions do not mitigate these shifts directly. Moreover, since $Y|X$-shifts often arise from spatial or temporal changes in real-world datasets, to the best of our knowledge, all datasets in SOTA benchmarks for tabular $Y|X$-shifts [2] include feature descriptions. Thus, we adopt the standard serialization scheme from [1], incorporating feature descriptions to explore the potential of using LLMs for tabular $Y|X$-shifts.
> - **We will clarify our limitations**. When feature descriptions are unavailable, our method can still be applied by serializing as "The feature 1 is {feature value}." However, we acknowledge that absence of feature descriptions can reduce benefits using LLMs. Thank you for your suggestions and we will discuss more about the limitations in the final version.
>
> [1] Hegselmann, Stefan, et al. "Tabllm: Few-shot classification of tabular data with large language models." International Conference on Artificial Intelligence and Statistics. PMLR, 2023.
>
> [2]  Liu, Jiashuo, et al. "On the need for a language describing distribution shifts: Illustrations on tabular datasets." Advances in Neural Information Processing Systems 36 (2023).
>
> ### **Q2. Comparison with TabTransformer**
> Thank you for highlighting this work. TabTransformer differs from our method in two ways:
> 1. TabTransformer does not require or utilize feature descriptions, making it applicable to other settings but losing the information provided by these descriptions.
>
> 2. It uses a late fusion scheme: embeddings of categorical features are concatenated with continuous features only after passing through the transformers and embedding layers. This limits interactions between feature types and does not heavily process continuous features, as we did with the LLM encoder, to extract richer representations.
>
> Given that popular $Y|X$-shift benchmarks include feature descriptions and TabTransformer differs fundamentally in architecture, comparing it to our method, even without feature descriptions, would not be entirely fair, as differences may arise from the architectures themselves. Nonetheless, TabTransformer is valuable for scenarios without feature names, and we will clarify our limitations in the next version.
>
> Further, we would like to clarify how we select the baselines.
> - **How we select baselines**: As for baselines, we (1) cover as many types of methods as possible, including basic models, deep models (TabPFN), robust methods (DROs), GBDTs, and in-context learning (2) select the SOTA methods (TabPFN and GBDTs)  according to previous benchmark papers [2, 3].
>
> [1] Hegselmann, Stefan, et al. "Tabllm: Few-shot classification of tabular data with large language models." International Conference on Artificial Intelligence and Statistics. PMLR, 2023.
>
> [2]  Liu, Jiashuo, et al. "On the need for a language describing distribution shifts: Illustrations on tabular datasets." Advances in Neural Information Processing Systems 36 (2023).
>
> [3] McElfresh, et al. “When Do Neural Nets Outperform Boosted Trees on Tabular Data?”. NeurIPS 2023.

---

> > ### Author Response · Authors · 2024-11-23
> >
> > Dear Reviewer pBwo,
> >
> > We are sincerely grateful for your time and efforts in the review process. Thank you so much for your support!
> >
> > We hope that our responses have been helpful. Given the limited time left in the discussion period, please let us know if there are any leftover concerns and if there is anything else we could do to address any further questions or comments. We are looking forward to your further feedback!
> >
> > Thank you!
> >
> > Paper Authors

---

### Author Response · Authors · 2024-11-21
**General Response 1: Our Contribution & Novelty (@MKos, @8GFo)**

We clarify our ***contribution and novelty*** in the following aspects:
- **Problem setting level**: In this paper, we tackle the challenging problem of tabular classification under $Y|X$ shifts. While the use of LLMs for classification and tabular data has been explored, LLMs’ performance under distribution shifts—particularly tabular $Y|X$ shifts—remains ***largely unexamined***. To the best of our knowledge, few, if any, studies have systematically investigated this problem [1], making our investigation a novel contribution in itself.
- **Benchmark level**:  We build a ***large-scale*** benchmark, encompassing 7,650 $Y|X$ shift settings—far surpassing the limited 169 $Y|X$ shift settings of prior SOTA tabular $Y|X$ shift studies [1]. This comprehensive evaluation, as supported by MKos in their point 1 of the strengths, highlights the robustness and generalizability of our approach. (See further discussion in dataset diversity).
- **Methodology level**: We are ***the first to thoroughly explore*** different ways to use LLM embeddings for tabular prediction, including linear probing, prefix tuning, finetuning and LoRA. We also explore different sources of additional information in helping with prediction.
- **Finding level**: We find that LLM can significantly improve model performance under $Y|X$ shifts for the first time in the literature to our best knowledge, even using standard LLM techniques. Our findings are also striking: using LLM embeddings, fine-tuning neural networks (NNs) with as few as 10–100 target samples can lead to substantial performance improvements in classification tasks. This is particularly surprising given the typically large number of parameters in NNs.
- **Paradigm shift in distribution shift research**: Our approach represents a significant paradigm shift in the field of distributional shift research. Traditionally, this field has focused on algorithmic design, investigating how methods such as distributionally robust optimization (DRO) or tree-based ensemble algorithms impact performance on the target domain. In contrast, our work emphasizes the transformative role of embeddings, ***opening an entirely new direction for tackling distributional shifts***.

[1]  Liu, Jiashuo, et al. "On the need for a language describing distribution shifts: Illustrations on tabular datasets." Advances in Neural Information Processing Systems 36 (2024).

---

> ### Author Response · Authors · 2024-11-21
> **General Response 2: Dataset Diversity (@Tko8, @ZwTt, @8GFo)**
>
> - **The diversity of distribution shift patterns ($Y|X$ shifts) should be prioritized over the diversity of dataset domains**, since the central challenge is how to evaluate whether a robust method performs well across most, if not all, unknown and varied level of $Y|X$ shift. A meaningful evaluation is only possible through diverse level of $Y|X$ shifts.
>
> - **Our evaluation covers diverse rough $Y|X$-shift levels**: We include ***7650 source-target shift pairs*** in total to reflect diverse $Y|X$-shift levels on three datasets, ACS Income, Public Coverage, and Mobility. The setup is ***much larger-scale*** than previous empirical studies. For instance, the WhyShift benchmark [1] has 169 settings, and TabLLM [2] only considers 9 settings.
> Moreover, each Y|X shift setting includes ~20K training samples and ~20K test samples, comparable to a typical dataset used in other tabular classification studies. For instance, TabLLM [2] considers 9 datasets, each up to 50K samples and 20 features.
>
> - **One more dataset from the public transportation domain**: We add the US accident dataset, which is to predict whether a traffic accident is severe or not given features like weather, visibility, etc. We consider data from 18 US states, and ***build another 18*17=306 $Y|X$-shift settings***. Our main findings still hold. As shown in the table below, LLM embedding alone (LLM | NN) already outperforms XGB and NNs using tabular features, and LLM | NN (finetuning) using very few target samples yields a notable ~1.5 pp improvement on average compare to LLM | NN across all $Y|X$-shift settings, the worst 200 Y|X shift settings, and the worst 100 $Y|X$-shift settings.
>
> | Methods                  | F1 Score All Settings | F1 Score Worst-200 Settings | F1 Score Worst-100 Settings |
> |--------------------------|-----------------------|-----------------------------|-----------------------------|
> | Tabular\|XGB             | 74.18% (±1.36%)       | 73.93% (± 1.71%)            | 74.04% (± 2.30%)            |
> | Tabular\|NN              | 75.29% (± 1.31%)      | 75.39% (±1.66%)             | 75.20% (± 2.23%)            |
> | **LLM\|NN**                  | **78.64%** (± 1.34%)      | **79.04%** (± 1.65%)            | **79.20%** (± 2.16%)            |
> | Tabular\|NN (finetuning) | 75.69% (± 0.64%)      | 75.73 (± 0.80%)             | 75.32% (±1.10%)             |
> | **LLM\|NN (finetuning)**     | **79.91%** (± 0.59%)      | **80.12%** (± 0.73%)             | **79.73%** (±1.04%)             |
>
>
> [1] Liu, Jiashuo, et al. "On the need for a language describing distribution shifts: Illustrations on tabular datasets." Advances in Neural Information Processing Systems 36 (2024).
>
> [2] Hegselmann, Stefan, et al. "Tabllm: Few-shot classification of tabular data with large language models." International Conference on Artificial Intelligence and Statistics. PMLR, 2023.

---

> > ### Author Response · Authors · 2024-11-21
> > **General Response 3: Test-Time Adaptation in the title (@Tko8, @ZwTt)**
> >
> > Thank you for your suggestions. As Tko8 and ZwTt correctly point out, our setting differs from the classical test-time adaptation, which typically relies solely on unlabeled target data. We use this title in our setting for two main reasons: (1) to emphasize the need to adapt the model to the target domain while allowing the use of various fine-tuning methods, such as vanilla fine-tuning, LoRA, and others; (2) addressing Y|X shifts must require labeled samples from the target domain, as we stated in Lines 035-039, so our setting is a generalization of the classical setting.
> >
> > We will include this discussion and clarify we need labeled target samples in the final version to clarify the differences.

---

### Comment · Area_Chair_eaFD · 2024-11-25

Dear Reviewers,

Thank you for your time and effort in reviewing for ICLR'25.

This is a gentle reminder to read and respond to the authors' rebuttals. Please submit your feedback in time. Thank you very much!

Best regards,

AC

---

### Meta-Review · Area_Chair_eaFD · 2024-12-20

**Metareview:**

This paper proposes a method to solve the Y|X shift problem in the tabular data domain using LLM embeddings. Under the LLM+NN model architecture, it uses concatenation to combine the domain information of LLM with the original tabular data information for improving prediction robustness and performance. The content of the paper is well written and intellectually challenging. However, the paper only conducts experiments on a limited number of datasets, which restricts the generalizability of the results. What’s more, the approach presented in the paper does not seem to be particularly novel or convincing and its theoretical results do not align well with the proposed method.   Therefore, I recommend the rejection of this paper.

**Additional Comments On Reviewer Discussion:**

After the rebuttal, the reviewers still maintain that the experimental section is insufficient and the method presented in the paper is unreasonable and does not seem to support the expected outcomes of the approach.

---

### Decision · Program_Chairs · 2025-01-22

Reject